# Enhanced Drug Skin Permeation by Azone-Mimicking Ionic Liquids: Effects of Fatty Acids Forming Ionic Liquids

**DOI:** 10.3390/pharmaceutics17010041

**Published:** 2024-12-30

**Authors:** Takeshi Oshizaka, Shunsuke Kodera, Rika Kawakubo, Issei Takeuchi, Kenji Mori, Kenji Sugibayashi

**Affiliations:** 1Faculty of Pharmaceutical Sciences, Josai International University, 1 Gumyo, Togane 283-8555, Chiba, Japan; pc19-030@jiu.ac.jp (S.K.); pc20-061@jiu.ac.jp (R.K.); itakeuchi@jiu.ac.jp (I.T.); kmori@jiu.ac.jp (K.M.); 2Graduate School of Pharmaceutical Sciences, Josai International University, 1 Gumyo, Togane 283-8555, Chiba, Japan; 3Faculty of Pharmacy and Pharmaceutical Sciences, Josai University, 1-1 Keyakidai, Sakado 350-0295, Saitama, Japan

**Keywords:** Azone, ionic liquid, skin permeation, chemical enhancer, ante-enhancer, fatty acid

## Abstract

**Background/Objectives**: Laurocapram (Azone) attracted attention 40 years ago as a compound with the highest skin-penetration-enhancing effect at that time; however, its development was shelved due to strong skin irritation. We had already prepared and tested an ante-enhancer (IL-Azone), an ionic liquid (IL) with a similar structure to Azone, consisting of ε-caprolactam and myristic acid, as an enhancer candidate that maintains the high skin-penetration-enhancing effect of Azone with low skin irritation. In the present study, fatty acids with different carbon numbers (caprylic acid: C8, capric acid: C10, lauric acid: C12, myristic acid: C14, and oleic acid: C18:1) were selected and used with ε-caprolactam to prepare various IL-Azones in the search for a more effective IL-Azone. **Methods**: Excised porcine skin was pretreated with each IL-Azone to assess the in vitro skin permeability of antipyrine (ANP) as a model penetrant. In addition, 1,3-butanediol was selected for the skin permeation test to confirm whether the effect of IL-Azone was due to fatty acids and if this effect differed depending on the concentration of IL-Azone applied. **Results**: The results obtained showed that C12 IL-Azone exerted the highest skin-penetration-enhancing effect, which was higher than Azone. On the other hand, many of the IL-Azones tested had a lower skin-penetration-enhancing effect. **Conclusions**: These results suggest the potential of C12 IL-Azone as a strong and useful penetration enhancer.

## 1. Introduction

Basic and applied research has been conducted on many pharmaceuticals and medicated cosmetics applied to the skin, with the goal of active drugs or ingredients exerting local or systemic effects, and many are already on the market. However, the skin, particularly the stratum corneum on the outermost layer, functions as the biggest skin barrier to protect the body against external bacteria and viruses and to prevent water evaporation from the body. Therefore, the skin may be difficult to penetrate for some active drugs or ingredients. To increase percutaneous absorption into the systemic circulation or penetration into the deep skin tissues, the high barrier function of the stratum corneum needs to be overcome. The skin permeability of a chemical compound is generally assessed by the physicochemical properties of the chemical as well as the barrier function of the skin barrier. Chemical compounds with the following physicochemical properties have high skin permeability: a molecular weight ≤500 Daltons [1] and moderate lipophilicity (log *n*-octanol/water partition coefficient of approximately 1–3) [2,3,4]. In other words, the skin permeability of high-molecular-weight and water-soluble chemical compounds is low. Since the 1980s, attention has been focused on increasing the skin permeability of active ingredients in topical medicines or medicated cosmetics. Chemical enhancement methods using penetration enhancers [5,6] and prodrugs [7], as well as physical enhancement methods using iontophoresis [8,9,10], electroporation [11], phonophoresis [12] and microneedles [13], have been attempted. However, many considerations are left for these physical enhancement methods, and they will take time before they can be put into practical use. For example, safety issues such as the malfunction of electrical equipment must be overcome for electroporation, iontophoresis, and ultrasound. In addition, microneedle technology is necessary to clarify the risk of infection in unsanitary environments and the risk of microneedles remaining in the skin if non-biodegradable microneedle materials are used. In contrast, chemical enhancement methods can be used if the toxicities of penetration enhancers contained in the formulation are cleared and safe in humans. Therefore, extensive research has been conducted on penetration enhancers because they may be easily incorporated into formulations. Ethanol, fatty acids, their esters, monoterpenes, and several surfactants have been evaluated as skin penetration enhancers [14,15,16,17]. Laurocapram (Azone), which was developed 40 years ago, is still known as a substance that exerts the markedly higher skin-penetration-enhancing effect rather than the abovementioned enhancers [5,18,19,20,21]; however, it is a strong skin irritant [22] and, thus, has never been applied to practical use. Thus, even chemical enhancement methods, which are thought to be easier to put into practice use than physical-penetration-enhancing methods, sometimes have problems with toxicity.

We focused on ionic liquids (ILs), known as “designer solvents”, with the aim of finding a compound that maintains the same penetration-enhancing effect as Azone, but with significantly reduced skin irritation [23,24,25,26,27,28]. ILs have also been reported in the field of pharmaceutical formulations, where an IL of choline bicarbonate and geranic acid was shown to increase insulin absorption from the intestinal tract [29]. ILs have been used to produce active ingredients for topical patches [30,31,32]. The mechanisms underlying the high percutaneous absorption of ILs have been investigated [25,26]. ILs have also been examined as solubilizers for drugs [33]. With reference to these studies, we discovered an IL that was similar in structure to Azone (IL-Azone), which may be defined as an ante-enhancer. IL-Azone exerts a skin-penetration-enhancing effect on drugs applied to the stratum corneum barrier, and is then converted to its original constituent compounds with low skin irritation at the viable epidermis and dermis under the stratum corneum, resulting in low skin irritation [34]. In addition, when IL-Azone and antipyrine (ANP), which was adopted as a model penetrant, were added to white petrolatum and a skin permeation test was performed, the skin-penetration-enhancing effect of IL-Azone was higher than that of Azone, while the storage stability of IL-Azone in white petrolatum was also high [35].

In the IL-Azone preparation, both ε-caprolactam and myristic acid were used. However, IL-Azone may also comprise other fatty acids. In the present study, IL-Azones were prepared using ε-caprolactam and various fatty acids with different carbon numbers, and their skin-penetration-enhancing effects were investigated by in vitro permeation experiments. Furthermore, the skin-penetration-enhancing effects of fatty acids alone as well as ILs composed of various fatty acids were tested because fatty acids themselves may exert skin-penetration- and gastrointestinal-absorption-enhancing effects [36,37].

## 2. Materials and Methods

### 2.1. Materials

Azone was purchased from Combi-Blocks Inc. (San Diego, CA, USA). ε-Caprolactam and myristic acid (C14) were obtained from Kanto Chemical Co., Inc. (Tokyo, Japan). ANP and propylene glycol were from Fujifilm Wako Pure Chemical Corporation (Osaka, Japan). Caprylic acid (C8), capric acid (C10), lauric acid (C12), oleic acid (C18:1), and 1,3-butanediol (1,3-BG) were purchased from Tokyo Chemical Industry Co., Ltd. (Tokyo, Japan). Methyl *p*-hydroxybenzoate as an internal standard was purchased from Tokyo Chemical Industry Co., Ltd. Other reagents and solvents were of HPLC quality or special grade and were used without purification.

### 2.2. Preparation of IL-Azones

The ε-caprolactam and one of the fatty acids (C8, C10, C12, C14, or C18:1) were mixed at a molar ratio of 1:1 and dissolved in methanol. During the vacuum distillation, the sample was placed in a water bath at 50 °C. After completely removing the solvent, vacuum distillation was continued for another 5 min to completely remove the methanol to obtain each IL-Azone. IL-Azones were also used to try to produce using C16 palmitic acid and C18 stearic acid, but these fatty acids were solids at 37 °C. Therefore, only C8-14 and C18:1 IL-Azones were tested in this study. Figure 1 shows the reaction scheme from ε-caprolactam and fatty acid to IL-Azone. Table 1 shows various IL-Azones prepared by the listed fatty acids used as raw materials.

### 2.3. Measurement of Melting Points Using a Differential Scanning Calorimeter (DSC)

DSC-60plus (Shimadzu Corp., Kyoto, Japan) was used to determine the melting points of each IL-Azone and its components, fatty acid and ε-caprolactam, under the following conditions: the measuring temperature range was 0 to 100 °C, the inside temperature was cooled by liquid nitrogen at the beginning of the measurement, the heating rate was 10 °C/min, the flow rate of N_2_ gas was 50 mL/min, and the sample amount was approximately 3 mg in an aluminum sealed cell.

### 2.4. Skin Penetration Enhancers

The skin penetration enhancers Azone and various IL-Azones were used as a pretreatment agent without the addition of other solvents. The enhancing effects of ε-caprolactam alone and various fatty acids alone were also evaluated as a pretreatment agent by suspending (or emulsifying) them in 1,3-BG. Figure 2 shows the preparation method of the suspended (or emulsified) solution (i.e., saturated solution) of ε-caprolactam and each fatty acid. ε-Caprolactam or each fatty acid was added to 1,3-BG in 20% the solubility, and the sample was immersed in a water bath at 37 °C overnight to prepare suspended (or emulsified) solutions. We confirmed that ε-caprolactam and some fatty acids (C12 and C14) were suspended in 1,3-BG, whereas other fatty acids (C8, C10, and C18:1) were found to separate into two layers when mixed in 1,3-BG at 37 °C. We applied these 1,3-BG solutions on the skin as a pretreatment agent.

The reason for using saturated solutions is that the effect of an enhancer is generally proportional to its thermodynamic activity [38]. Therefore, we examined IL-Azones in suspended or emulsified solutions (i.e., saturated solution), which have the maximum thermodynamic activity [36]. Furthermore, C12 alone and the physical mixture of C12 and ε-caprolactam were suspended in 1,3-BG and used as a pretreatment agent to examine their skin-penetration-enhancing effects. Separately, in the evaluation test of C12 concentration in IL-Azone, IL-Azone was dissolved in 1,3-BG at C12 concentrations of 5, 10, and 15% and was used as a pretreatment agent to confirm whether a C12 dose-dependent penetration-enhancing effect was exerted.

### 2.5. In Vitro Skin Permeation Experiments

The frozen ears of edible male and female pigs (LWD, 5–6 months old) were obtained from the Central Research Institute for Feed & Livestock (National Federation of Agricultural Cooperative Associations, Tsuchiura, Ibaraki, Japan). The skin was stored at −25 °C until permeation experiments.

Frozen pig ears were thawed at room temperature, and the hairs on the ear were carefully cut with scissors. Ear skin was rinsed with purified water and excised from the pig ears. Debris and excess fat were trimmed off to obtain the test skin. The test skin pieces obtained were set in horizontal 2-chamber diffusion cells (permeation area: 0.95 cm^2^). Approximately 3.0 mL of each skin penetration enhancer or 1,3-BG was added to the donor cell and approximately 3.0 mL saline was added to the receiver cell, and the pretreatment was performed for 16 h. After the pretreatment, the penetration enhancer was removed from the donor side and wiped with a cotton pad soaked with purified water. ANP in purified water (1.0 mg/mL) was added to the donor cells to start the permeation experiment. The whole cell was maintained at 37 °C during the experiment. The donor and receiver solutions were stirred with a magnetic stirrer during the experiment.

Receiver liquid samples (1.0 mL) were collected periodically, and the same amount of saline was returned to maintain the sink condition of receiver solutions. Receiver samples containing ANP were measured by HPLC.

### 2.6. Analytical Methods of ANP

ANP was measured using HPLC. The HPLC system consists of a pump (LC-10AD_VP_; Shimadzu, Kyoto, Japan), UV detector (SPD-10A_VP_; Shimadzu), system controller (SCL-10A_VP_; Shimadzu), column oven (CTO-10A; Shimadzu) and an auto-injector (SIL-10AF; Shimadzu). The sample containing ANP (500 μL) was mixed with the same volume of methanol solution containing methyl *p*-hydroxybenzoate as an internal standard (10 μg/mL) and centrifuged at 15,000 rpm and 4 °C for 5 min. The supernatant obtained (20 μL) was injected into an HPLC system. The column was TSKgel ODS-80Ts QA (TOSOH, Tokyo, Japan), maintained at 40 °C, the mobile phase was 0.1% phosphoric acid in water:acetonitrile = 70:30 containing 5.0 mM tetrabutylammonium hydrogen sulfate, and the flow rate was adjusted to 1.0 mL/min. ANP was detected at UV 245 nm.

### 2.7. Statistics

Statistics for the skin permeation data were obtained by paired *t*-test.

## 3. Results

Figure 3a,b shows the DSC charts for each fatty acid and ε-caprolactam, and each IL-Azone, respectively. The endothermic peaks of C8, C10, C12, C14, C18:1, and ε-caprolactam were found at 16.2, 32.6, 45.5, 57.3, 11.9, and 71.6 °C, respectively (Figure 3a). Since their melting points are known to 16.7, 31.6, 45.0, 54.0, 13.4, and 70.0 °C, respectively, these endothermic peaks must be due to their melting. On the other hand, in the DSC charts for each IL-Azone, no endothermic peaks were attributed to the melting points of their constituent fatty acids and ε-caprolactam (Figure 3b), suggesting that IL-Azone was successfully produced.

Endothermic peaks at 13.9 °C and 25.1 °C for C12 IL-Azone and C14 IL-Azone, respectively (Figure 3b), must be attributed to the melting point for the ILs. The melting points for C8, C10, and C18:1 IL-Azone are probably below 0 °C.

Figure 4 shows the cumulative amount of ANP that permeated through skin pretreated with Azone or each IL-Azone over time. Saline was also tested as a control. The amount of ANP that permeated skin pretreated with C12 IL-Azone over 24 h was the highest, followed by C18:1 > C10 > C14 > C8 IL-Azone. The permeation of ANP was significantly higher through skin pretreated with these IL-Azones than through that pretreated with saline (*p* < 0.01 for C10, C12, and C18:1 IL-Azone, and *p* < 0.05 for C8 and C14 IL-Azone). The permeation of ANP with Azone or C12 IL-Azone pre-treatment was also significantly higher than in C18:1 IL-Azone, C10 IL-Azone, C14 IL-Azone, and C8 IL-Azone pre-treatment (*p* < 0.01).

Furthermore, the cumulative amount of ANP that permeated through skin pretreated with C12 IL-Azone was significantly higher than that through skin pretreated with Azone (*p* < 0.05).

Figure 5 shows the permeation of ANP through skin pretreated with each component of IL-Azone (each fatty acid or ε-caprolactam) suspended in 1,3-BG over time. The cumulative amount of ANP that permeated through skin pretreated with each fatty acid suspended in 1,3-BG was significantly higher than that without the fatty acid (1,3-BG alone) (*p* < 0.01 for C8, C10, and C18:1, and *p* < 0.05 for C12 and C14). On the other hand, no significant difference was observed between the cumulative amount of ANP that permeated through skin pretreated with the ε-caprolactam suspension in 1,3-BG and 1,3-BG alone. Therefore, fatty acids exerted a skin-penetration-enhancing effect, whereas ε-caprolactam did not.

Table 2 compares the penetration-enhancing ratios of ANP by various IL-Azones with saline, and various fatty acids or ε-caprolactam suspensions with 1,3-BG. Penetration-enhancing ratios were compared based on the cumulative amount of ANP that permeated over 24 h. The penetration-enhancing ratios of C8, C10, and C18:1 IL-Azones were lower or similar to the corresponding fatty acid-saturated solution. On the other hand, the penetration-enhancing ratios of C12 and C14 IL-Azones were higher than the corresponding fatty acid suspensions.

We then investigated whether a simple physical mixture of two components of IL-Azone exerted an enhancing effect. We only tested C12 because C12 IL-Azone was found to exert the highest penetration-enhancing effect. Figure 6 shows the skin permeation of ANP when ε-caprolactam and C12 were simply suspended in 1,3-BG. C12 suspension data in 1,3-BG (shown in Figure 5) were added for comparison. This simple mixture of ε-caprolactam and C12 showed a similar permeation profile to that of the single C12 suspension.

Figure 7a shows ANP permeation data for skin pretreated with IL-Azone dissolved in 1,3-BG at concentrations of 5, 10, and 15%. Figure 7b shows the relationship between the permeation amount over 24 h and the concentration of IL-Azone. No significant differences were observed between 5% IL-Azone in 1,3-BG and 1,3-BG alone. The skin-penetration-enhancing effects of 10 and 15% IL-Azone in 1,3-BG were significantly high (*p* < 0.05) and dependent on the IL-Azone dose applied.

## 4. Discussion

Based on the DSC charts in Figure 3, the disappearance of the endothermic peaks attributed to the melting points of individual components after IL formation suggests that ILs were successfully formed.

The cumulative amount of ANP that permeated through skin pretreated with various IL-Azones was higher than that with physiological saline, regardless of the fatty acid used as the raw material (C8, C10, C12, C14, and C18:1). Among these IL-Azones, the highest permeation of ANP occurred with C12 IL-Azone, exceeding even that of Azone. This result suggests that the skin-penetration-enhancing effect of IL-Azone varies with the carbon chain length of the fatty acid component, with C12 IL-Azone having the greatest impact. It is known that Azone shows a skin-penetration-enhancing effect by acting in the stratum corneum. Many studies have been conducted to clarify its penetration-enhancing mechanism, revealing that Azone exerts its penetration-enhancing effect by fluctuating the intercellular lipids or creating holes in the intercellular lipids of the stratum corneum. IL-Azones prepared in this study act in the stratum corneum in the same way as Azone. Further research into the mechanism of IL-Azone will become a more effective and safe penetration enhancer.

Fatty acids alone have been shown to enhance the skin permeation of chemical compounds [36]. Ideally, when comparing the skin-penetration-enhancing rates of fatty acids and IL-Azone, the fatty acid concentration in IL-Azone needs to match that of a pure fatty acid solution. However, since IL-Azones contain extremely high concentrations of fatty acids, and fatty acids alone have limited solubility in 1,3-BG, ANP permeability was tested using skin pretreated with fatty acids suspended or emulsified in 1,3-BG. As previously described, the skin-penetration-enhancing effect is affected by the thermodynamic activity of the enhancer [38]. In other words, fatty-acid-saturated solutions typically exhibit the highest enhancing effects. The cumulative amount of ANP that permeated through skin pretreated with fatty acids saturated in 1,3-BG was higher than that of 1,3-BG alone for all fatty acids tested. Furthermore, the cumulative permeation of ANP with ε-caprolactam suspended in 1,3-BG was similar to that of 1,3-BG alone, indicating that ε-caprolactam had no enhancing effect.

In comparisons of the penetration-enhancing effects of IL-Azones and their corresponding fatty acid solutions, saturated solutions of C8, C10, and C18:1 fatty acids alone exerted higher effects than the corresponding IL-Azones. Conversely, C12 and C14 IL-Azones achieved greater enhancement than the corresponding fatty acids. Therefore, converting C8, C10, and C18:1 into IL-Azone is unnecessary due to the labor involved in the conversion process. On the other hand, the penetration-enhancing effects of C12 and C14 IL-Azones were significantly increased when these fatty acids were converted into IL-Azones. Specifically, the enhancement with C12 IL-Azone was 4.3-fold higher than that with the C12 fatty acid suspension. Additionally, skin permeability from a physical mixture of ε-caprolactam and C12 in 1,3-BG was similar to that of the C12 suspension alone. Therefore, a high penetration-enhancing effect may be achieved by converting ε-caprolactam and C12 into IL-Azone. Moreover, a penetration-enhancing effect was not observed at 5% IL-Azone, while significant enhancement occurred at concentrations of 10 and 15%, indicating that IL-Azone needs to be used at a concentration of 10% or higher.

The present results demonstrate that C12 IL-Azone exerted a higher skin-penetration-enhancing effect than Azone and that converting Azone into IL-Azone may reduce skin irritation [34]. The development of advanced enhancers using IL technology holds great potential for future drug delivery systems in pharmaceutical applications.

## 5. Conclusions

Azone, examined in the 1980s, exerted an exceptionally high skin-penetration-enhancing effect, leading to the synthesis of many derivatives over the years [39]. However, none of these derivatives have been successfully applied in practice due to significant skin irritation. In our previous research, which focused on IL technology using ε-caprolactam and C14, we developed a skin penetration enhancer with a similar skin-penetration-enhancing effect to Azone. Although C14 IL-Azone was less irritating than Azone, its penetration-enhancing effect was lower [34]. In the present study, by switching from C14 IL-Azone to C12 IL-Azone, we achieved a skin-penetration-enhancing effect that surpassed that of Azone. In addition, we have already reported that neat C14 IL-Azone has lower skin irritation than Azone [34]. In practical use, IL-Azone will be rarely used in its original form; so, a primary skin test was conducted with 3% C14 IL-Azone in 1,3-BG, and the results showed almost no skin irritation [34]. C12 IL-Azone, which had a higher permeation-promoting effect than Azone in the present study, may also be expected to present lower skin irritation than Azone. Skin irritation tests will need to be conducted for C12 IL-Azone in the future. In addition, the mechanism of skin-penetration-enhancing effect of IL-Azone needs to be tested. Since ε-caprolactam has not been approved as a drug or a cosmetic additive, many safety tests will be required. Thus, it is desirable to create a new IL-Azone using a different substance than ε-caprolactam and examine its skin-permeation-promoting effect in the future.

Additionally, Azone takes approximately 10 h to exert its full skin-penetration-enhancing effect. Therefore, in the present study, we applied a 16 h pretreatment with various enhancers before conducting the skin permeation experiments. Our previous findings showed that IL-Azone, consisting of ε-caprolactam and C14, when formulated in white petrolatum, achieved a high penetration-enhancing effect within a short period, surpassing Azone’s performance [35]. In the future, we plan to investigate the formulation of IL-Azone, composed of ε-caprolactam and C12, in white petrolatum in order to further optimize its efficacy.

## Figures and Tables

**Figure 1 pharmaceutics-17-00041-f001:**
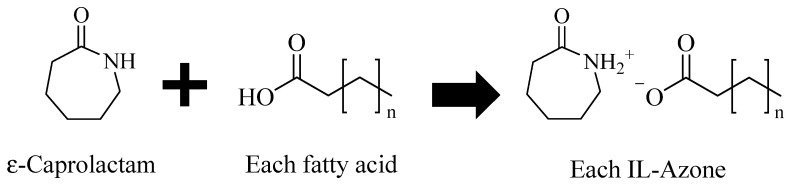
The reaction scheme to IL-Azone.

**Figure 2 pharmaceutics-17-00041-f002:**
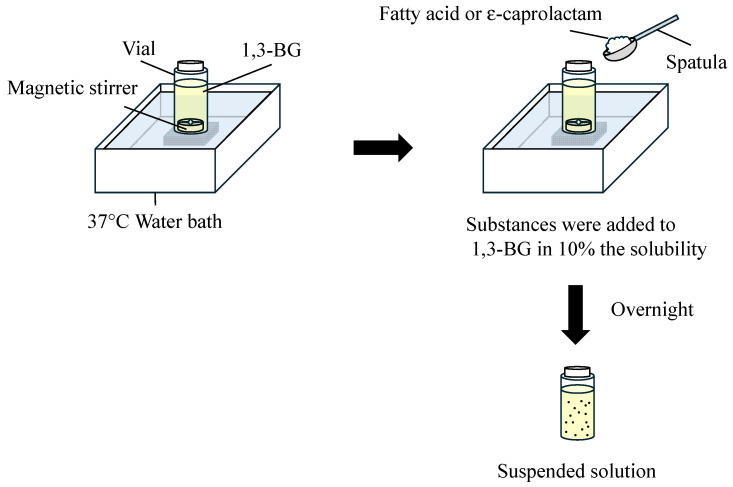
Preparation methods of suspended or emulsified solution (i.e., saturated solution) containing various fatty acids or ε-caprolactam.

**Figure 3 pharmaceutics-17-00041-f003:**
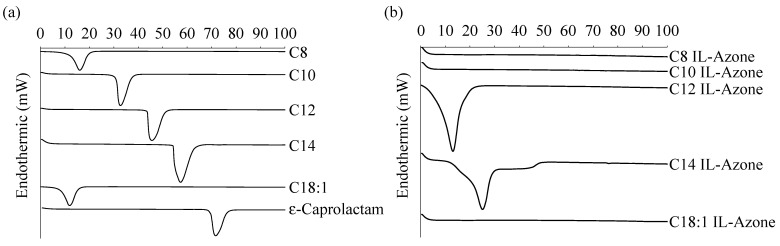
DSC charts of IL-Azone components (**a**) and each IL-Azone (**b**).

**Figure 4 pharmaceutics-17-00041-f004:**
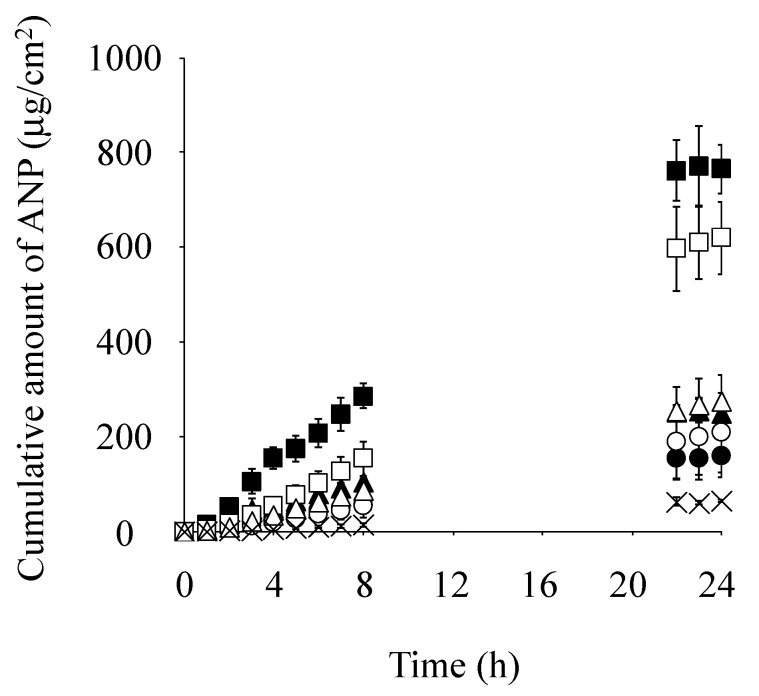
Pretreatment effects of Azone and each IL-Azone on the cumulative amount of ANP permeating through the skin over 24 h. Symbols: control (saline; ×), IL-Azone (C8; ⬤, C10; ▲, C12; ■, C14; ◯, and C18:1; △), and Azone (☐). Each data point represents the mean ± S.D. (*n* = 3–4).

**Figure 5 pharmaceutics-17-00041-f005:**
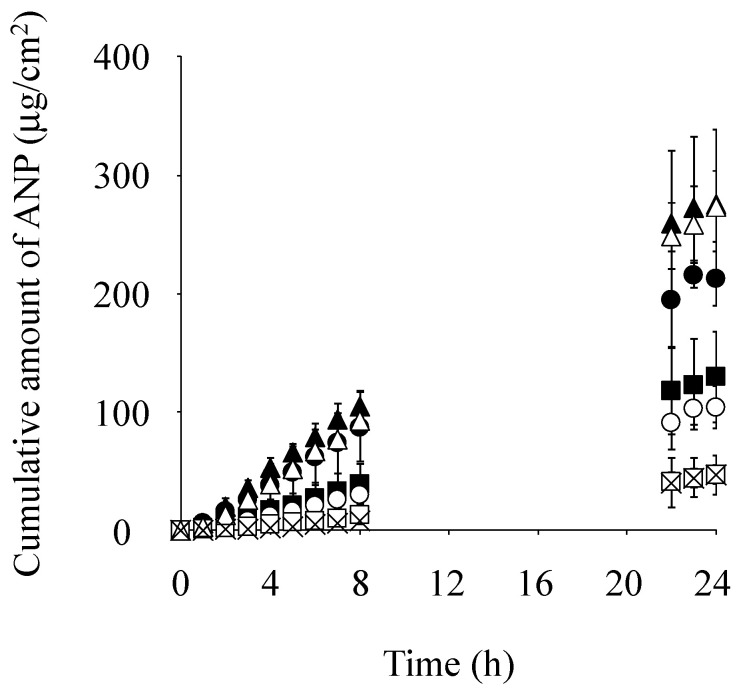
Pretreatment effects of each fatty acid or ε-caprolactam suspended in 1,3-BG on the cumulative amount of ANP that permeated through the skin over 24 h. Symbols: control (control, 1,3-BG alone; ×), each fatty acid or ε-caprolactam suspended in 1,3-BG (C8; ⬤, C10; ▲, C12; ■, C14; ◯, C18:1; △, and ε-caprolactam; ☐). Each data point represents the mean ± S.D. (*n* = 3–6).

**Figure 6 pharmaceutics-17-00041-f006:**
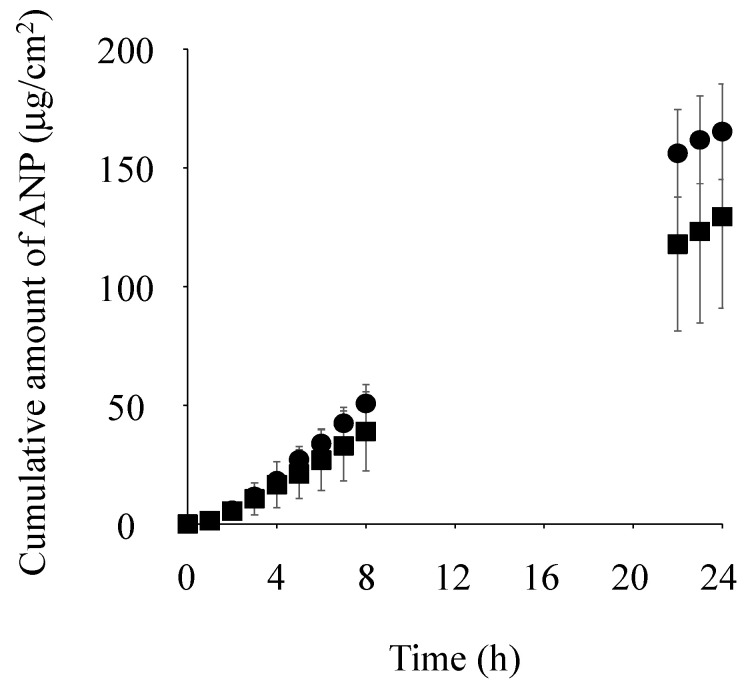
Pretreatment effects of a simple physical mixture of C12 and ε-caprolactam suspended in 1,3-BG on the time course of the skin permeation of ANP. Symbols: suspension solution (C12 and ε-caprolactam; ⬤) and C12 alone (■) (the same data as in Figure 5). Data points represent the mean ± S.D. (*n* = 3).

**Figure 7 pharmaceutics-17-00041-f007:**
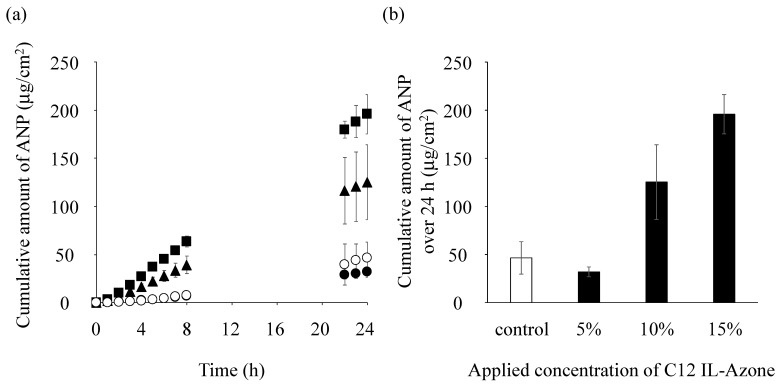
Pretreatment effects of C12 IL-Azone in 1,3-BG at a C12 concentration of 5, 10, or 15% on the skin permeation of ANP. (**a**) Time course of the skin permeation of ANP. Symbols: control (1,3-BG alone; ◯), 5% IL-Azone of C12 (⬤), 10% IL-Azone of C12 (▲), 15% IL-Azone of C12 (■). (**b**) Relationship between the cumulative amount of ANP that permeated through skin over 24 h after the application of ANP with C12 IL-Azone. Each data point and column represent the mean ± S.D. (*n* = 3).

**Table 1 pharmaceutics-17-00041-t001:** List of various IL-Azones and fatty acids used.

Name of Fatty Acid	Number of C Atoms	Ionic Liquid	Designation
Caprylic acid	8	2-oxoazepan-1-ium octanoate	C8 IL-Azone
Capric acid	10	2-oxoazepan-1-ium decanoate	C10 IL-Azone
Lauric acid	12	2-oxoazepan-1-ium dodecanoate	C12 IL-Azone
Myristic acid	14	2-oxoazepan-1-ium tetradecanoate	C14 IL-Azone
Oleic acid	18:1	2-oxoazepan-1-ium stearate	C18:1 IL-Azone

**Table 2 pharmaceutics-17-00041-t002:** Skin-penetration-enhancing effects of Azone, IL-Azones, and fatty acids *.

	Azone or Each IL-Azone vs. 0.9% NaCl **	Each Fatty Acid-Saturated Solution vs. 1,3-BG Alone ***	Each Fatty Acid-Saturated Solution vs. 0.9% NaCl **^,^***
Azone	9.6	-	-
C8	2.5	4.6	3.3
C10	3.9	5.9	4.3
C12	11.9	2.8	2.0
C14	3.3	2.2	1.6
C18:1	4.3	5.9	4.3
ε-caprolactam	-	1.0	0.7

* Each value shows the ratio of the cumulative amount of ANP that permeated through the skin over 24 h for each treatment against that for 0.9% NaCl or 1,3-BG alone. **^,^*** The time courses of the skin permeation of ANP are shown in Figure 4 and Figure 5.

## Data Availability

Data are contained within the main article.

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
