# Peer review of "Enhanced Drug Skin Permeation by Azone-Mimicking Ionic Liquids: Effects of Fatty Acids Forming Ionic Liquids"

_pharmaceutics, 2024, doi:10.3390/pharmaceutics17010041_

Round 1
Reviewer 1 Report
Comments and Suggestions for Authors
In this article, the authors prepared an ante-enhancer (IL- 16 Azone), an ionic liquid (IL) with a similar structure to Azone, consisting of a cationic substance (e- caprolactam) and anionic substance (myristic acid), The results showed that C12 IL-Azone exerted the highest skin penetration-enhancing effect, which was higher than Azone. The study falls within the purview of the Pharmaceutics. I recommend accepting this manuscript with major revisions. Here are the comments:
1. Is there any evidence to prove they are ionic liquid? Can you add DSC results? FTIR results? and other prove results?
2. The mechanism of Azone is dissolve the lipids in the skin, how about the mechanism of your ionic liquid?
3. The ionic liquid can dissolve in 13BG, how about the physical mixture in figure 3?
4. In table 1, can we all VS 0.9% NaCl?
5. How about the stability of the C10 suspension in 1,3 BG?
6. Is there any significant difference in figure 4?
7. What is the difference with the reference 35?
Reviewer 2 Report
Comments and Suggestions for Authors
Article report
Article: Enhanced Drug Skin Permeation by Azone-Mimic Ionic Liquids: Effects of Fatty Acids, a Constituent of Ionic Liquids
Authors: Takeshi Oshizaka, Shunsuke Kodera, Rika Kawakubo, Issei Takeuchi, KenjiMori and Kenji Sugibayashi
• A brief summary
The authors examine a group of compounds from the class of ionic liquids for the
presence of properties of local delivery of medical preparations using the property of ionic liquids to overcome the protective barrier function of the skin - the stratum corneum. Skin diseases such as lupus, cancer, psoriasis, and hyperhidrosis can potentially be treated effectively using ionic liquids as supplier of essential drugs. Previously, the authors found an ionic liquid С12 IL-Azone with low skin irritation that was successful for the stated purpose, described in reference [24]. In order to search for other ionic liquids with enhanced drug skin permeation properties, this article presents a group of ionic liquids obtained from epsilon-caprolactam and fatty carbonic acids. It turned out that as a result of this work, no better ILs than C12 IL-Azone were found.
The article provides sufficient references to modern literature describing the area of ​​research.
• General concept comments
There are a number of terminological and experimental inaccuracies.
The production of azone-mimic ionic liquids in the form of quaternary ammonium salts occurs through the interaction of epsilon-capolactam and fatty carboxylic acids. The epsilon-capolactam fragment becomes the cationic center and the anionic component is formed from the acid in a chemical reaction. In order for the authors' results to be reproducible, the newly obtained compounds ‒ quaternary ammonium salts need to be characterized.
Quaternary ammonium salts must be characterized by boiling points if they are liquids or melting points if they are crystalline substances. And, in addition, it is necessary to indicate the yield (%), mobility index (Rf), elemental analysis and temperature and reaction time. Also, new compounds must be characterized by spectral methods: IR spectroscopy, 1H and 13C NMR spectroscopy. X-ray structural analysis is also not superfluous in the case of crystalline substances.
The article lacks a reaction scheme, which complicates the perception of the material. The authors also use imprecise terminology. It is necessary to provide a reaction scheme and make distinctions in the names of the reagents and the resulting ionic liquids; perhaps these thoughts can be presented in the following view:

|
Name of carboxylic acid |
Number of C-atoms |
Ionic Liquid |
Designation |
|
Caprylic acid |
8 |
2-oxoazepan-1-ium octanoate |
C8 IL-Azone |
|
Capric acid |
10 |
2-oxoazepan-1-ium decanoate |
C10 IL-Azone |
|
Lauric acid |
12 |
2-oxoazepan-1-ium dodecanoate |
C12 IL-Azone |
|
Myristic acid |
14 |
2-oxoazepan-1-ium tetradecanoate |
C14 IL-Azone |
|
Oleic acid |
18 |
2-oxoazepan-1-ium stearate |
C18 IL-Azone |
The text of the article must contain the correct name of the ionic liquids. In this regard, the title of the article should be formulated more precisely. For example: Enhanced Drug Skin Permeation by Azone-Mimic Ionic Liquids: Effects of Fatty Acids Forming Ionic Liquids.
• Specific comments. (Line 103). The authors in the section 2.1 Preparation of IL-Azones call epsilon-caprolactam a cationic substance. This is incorrect, since it is not a cation, but an uncharged compound. Fatty acids are also called anionic substances. It should be noted that the fatty acids used in the work become anions when the reaction of formation of ionic liquids in the form of quaternary ammonium salts occurs.
· The manuscript is clear, relevant for the field and presented in a well-structured manner.
· The cited references mostly recent publications (within the last 5 years) and relevant. The
authors refer to their own works, but without excessive self-citation.
· The manuscript scientifically sound. The experimental hypothesis, consisting of the
search for consonance in the previously discovered properties of Azone and easily obtained ILs based on epsilon-caprolactam and fatty acids, is justified and has found its embodiment in the compound C12 ILAzone discovered by the authors.
• I think that the article lacks illustrations and as an option above I offered my version. In addition, it might not be superfluous to have a visual representation of the experiment with obtaining suspensions and presenting its sequence. As for the statistical methods of processing the experiment, the authors indicate: Statistics for the skin permeation data were done by paired t-test.
· I believe that the conclusions are consistent with the evidence and arguments presented.
· The article lacks an ethics statement as it is inappropriate due to the lack of animal or human studies.
Rating the Manuscript
· Novelty: The question of the article is original and well-defined. The results provide an advancement of the current knowledge.
· Scope: The work fit the journal scope.
· Significance: The results are not interpreted accurately enough because there is a gap in the physicochemical and spectral data.
· Quality: I think the article lacks data to identify new compounds.
• Scientific Soundness: The lack of characterization of the new compounds obtained in this work leads to the conclusion that the study is not correctly designed and technically sound.
It can be said that the conclusions put forward have a probability of being reliable, since the work complies with the statistical approach and there are data on the variation of the concentration of the IL with Enhanced Drug Skin Permeation property and the variation of the conditions of the experiment.
In order for other experimenters to be able to reproduce the results, it is necessary to specify the experimental conditions in detail and characterize the new compounds obtained by the accepted methods.
• Interest to the Readers: I think that the Pharmaceutics Journal has its own circle of readers who will be interested in this article.
• Overall Merit: The authors are exploring a fairly relevant topic ‒ transdermal drug delivery, bypassing injection or oral administration for safer effects on internal organs. For this reason, the work is of interest. The authors empirically discovered an active ionic liquid C12 IL-Azone with a less irritating effect and greater efficiency, compared to Azone. These data may have practical application in medicine and cosmetics after studying the toxic properties of the promising compound.
· English Level: The English language appropriate and understandable.
· Overall Recommendation:
Reconsider after Major Revisions.

Round 2
Reviewer 1 Report
Comments and Suggestions for Authors
I have no comments now.
Reviewer 2 Report
Comments and Suggestions for Authors
Comments to the revised article: Enhanced Drug Skin Permeation by Azone-Mimic Ionic Liquids: Effects of Fatty Acids, a Constituent of Ionic Liquids
Authors: Takeshi Oshizaka * , Shunsuke Kodera , Rika Kawakubo , Issei Takeuchi , Kenji Mori , Kenji Sugibayashi *
Section: Drug Delivery and Controlled Release
Special Issue: Transdermal Delivery of Low-Molecular-Weight Drugs and New Modality of Drugs: Recent Innovation to Penetration Enhancement Techniques
1. The authors have revised the article in accordance with the reviewer's wishes. In my opinion, the article has become more convincing and clear.
2. In addition, and very importantly, in the revised text the authors presented a research method specific to the field of ionic liquids in section 2.2. - Measurement of melting points using a differenaial scanning calorimeter (DSC) DSC charts. This part was not in the previous text.
3. The Originality/Novelty, Significance of Content, and Overall Merit indicators are assessed as “average” indicators due to the fact that the authors had previously found the most active IL from the presented series in [24]; in the article, the range of objects that serve as a background for the already found successful IL sample, C12 IL, is expanded.
4. The paper can be accepted without any further changes.
